# Enhancing bacteriophage therapeutics through in situ production and release of heterologous antimicrobial effectors

Jiemin Du [1,3], Susanne Meile [1,3], Jasmin Baggenstos[1], Tobias Jäggi[1], Pietro Piffaretti[1], Laura Hunold[1], Cassandra I. Matter[1], Lorenz Leitner[2], Thomas M. Kessler [2], Martin J. Loessner [1], Samuel Kilcher [1] ✉ & Matthew Dunne [1] ✉

Bacteriophages operate via pathogen-specific mechanisms of action distinct from conventional, broad-spectrum antibiotics and are emerging as promising alternative antimicrobials. However, phage-mediated killing is often limited by bacterial resistance development. Here, we engineer phages for target-specific effector gene delivery and host-dependent production of colicin-like bacteriocins and cell wall hydrolases. Using urinary tract infection (UTI) as a model, we show how heterologous effector phage therapeutics (HEPTs) suppress resistance and improve uropathogen killing by dual phage- and effector-mediated targeting. Moreover, we designed HEPTs to control polymicrobial uropathogen communities through production of effectors with cross-genus activity. Using phage-based companion diagnostics, we identified potential HEPT responder patients and treated their urine ex vivo. Compared to wild-type phage, a colicin E7-producing HEPT demonstrated superior control of patient *E. coli* bacteriuria. Arming phages with heterologous effectors paves the way for successful UTI treatment and represents a versatile tool to enhance and adapt phage-based precision antimicrobials.

Currently, conventional, small-molecule antibiotics with broad target specificity are the most effective treatments against bacterial infections. However, the global emergence and spread of antimicrobial resistance (AMR)[1], as well as adverse effects caused by antibiotic-induced microbiome dysbiosis, highlight the need for novel and more pathogen-specific antimicrobial interventions[2,3]. Bacteriophages (phages), bacteriocins, synthetic antimicrobial peptides, and target-specific cell wall hydrolases (e.g., phage-derived endolysins) are currently being developed as precision antimicrobials[4]. Among these, phages are highly promising because of their ubiquity, pathogen specificity, and ability to self-replicate[5,6]. Although the killing of host bacteria by phages is largely independent of the host drug-resistance profile, treatment with phages often fails to inactivate all bacterial cells within a target population. This can be due to phage tolerance[7] or

resistance mechanisms that bacteria employ to counteract viral predation, including the production of extracellular matrices, mutation or reduced expression of phage receptors, adaptive CRISPR-Cas immunity, restriction/modification systems, abortive infection systems, and a growing number of other resistance mechanisms described in the literature[8,9].

Recent advances in CRISPR-Cas technology and synthetic biology have enabled the rapid modification of phage genomes beyond model phages (such as T4, T7, or lambda) to include therapeutic phage candidates that are typically less well-studied[10]. As a result, engineering has been applied (i) to adapt phage tropism through directed receptor binding protein modification[11–14], (ii) to construct sequence-specific antimicrobials through phage-mediated, pathogen-specific delivery of programmed CRISPR-Cas modules[15–17], (iii) to deliver toxic proteins as

[1]Institute of Food Nutrition and Health, ETH Zurich, Zurich, Switzerland. [2]Department of Neuro-Urology, Balgrist University Hospital, University of Zurich, Zurich, Switzerland. [3]These authors contributed equally: Jiemin Du, Susanne Meile. ✉e-mail: samuel.kilcher@hest.ethz.ch; matthew.dunne@hest.ethz.ch

genetic payloads[18], (iv) to develop rapid phage-based (companion) diagnostics through the delivery of reporter genes (reporter phages)[10,19,20], and (v) to optimize therapeutic phages for experimental therapy[21,22].

In this study, we demonstrate how diverse phages can be engineered to encode bacteriocins and cell wall hydrolases as antimicrobial effector genes, a concept we coin heterologous effector phage therapeutics (HEPTs). Here, effector genes are expressed during infection and their products released upon host cell lysis to function as secondary pathogen-specific antimicrobials, thereby complementing and enhancing phage-mediated killing.

## Results and discussion

As a model system, we focused on developing HEPTs as precision antimicrobials against UTI pathogens (concept: Fig. 1a). UTIs are among the most common community-acquired and healthcare-associated microbial infections in all age groups and a major public health concern, resulting in annual healthcare costs exceeding 2.8 billion dollars in the US alone[23,24]. While the most prevalent causative agent of UTIs is *Escherichia coli*, the microbial etiology is complex and can involve a wide range of Gram-negative or Gram-positive bacteria and certain fungi[25]. An analysis of 339 isolates acquired from 227 incidents of UTI during 2020 in Zurich, Switzerland (the Zurich

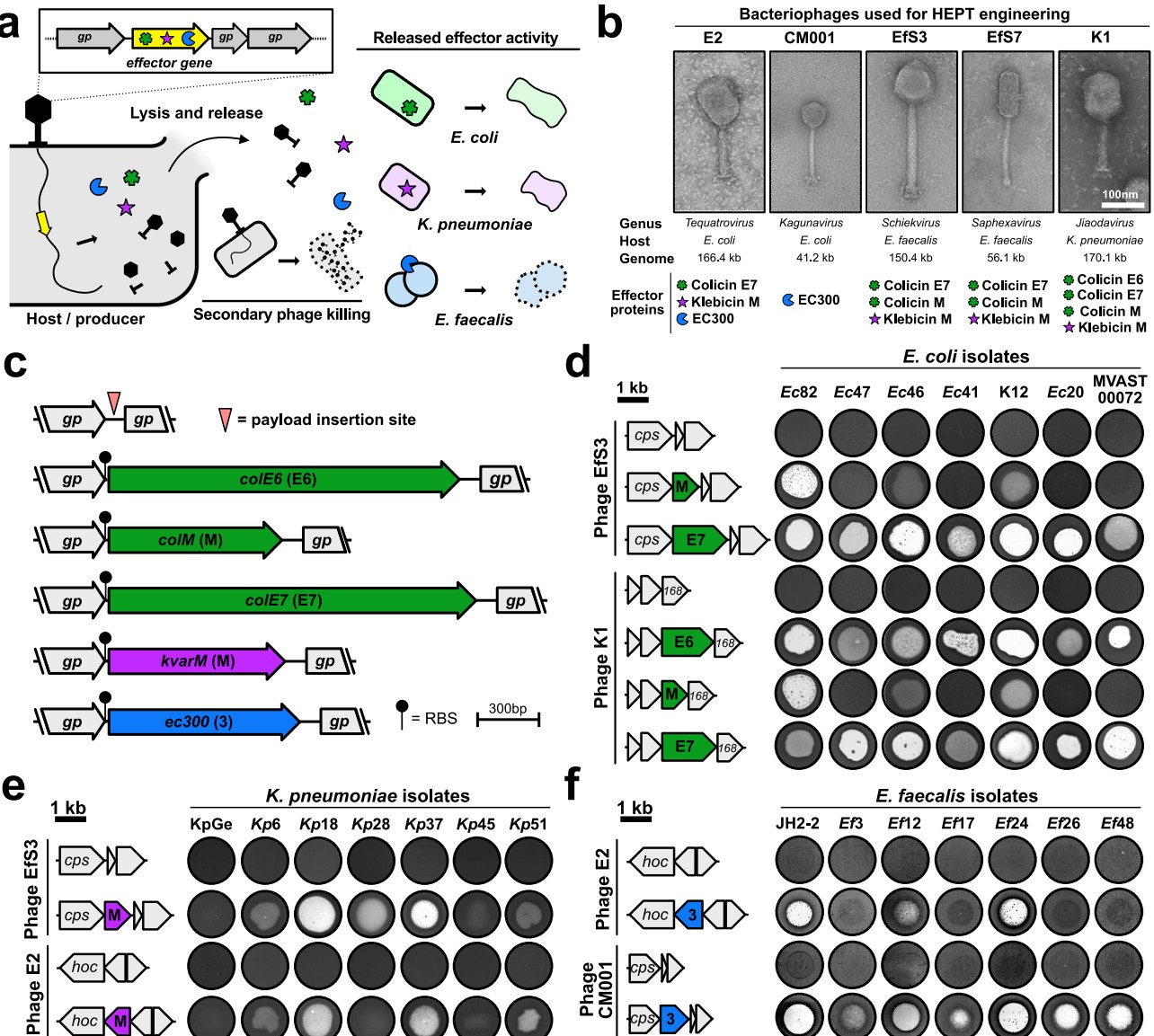

**Fig. 1 | HEPT construction and assessment of payload activities. a** Heterologous effector phage therapeutics (HEPTs) enable pathogen-specific gene delivery and production of antimicrobial effectors. Upon phage-induced host cell lysis, effectors are released alongside progeny virions to exert a secondary antimicrobial activity against defined bacterial targets. HEPTs were designed against the uropathogens *E. coli* (light green), *K. pneumoniae* (light purple), and *E. faecalis* (light blue). **b** Five phages were employed as HEPT scaffolds to integrate colicins (M, E6, and E7; square star), klebicin M (star), or the *E. faecalis*-specific cell wall hydrolase EC300 (Pac-Man). The colour of the effector symbol matches the target organism. **c** Genes encoding for colicin-like bacteriocins or EC300 were codon-optimized for the producing host-organism and integrated within the structural gene cluster of the corresponding phage scaffold alongside a strong ribosomal binding site (RBS) to mediate phage late promoter-driven effector expression. **d–f** Cross-genus antimicrobial activity of crude WT phage or HEPT lysates (~10⁹–10¹⁰ PFU/mL) was tested against clinical uropathogen isolates using the spot-on-the-lawn method (full lists provided in Supplementary Table 2 and Supplementary Data 1). Clear zone formation at the site of HEPT lysate spot indicates bacterial susceptibility to the corresponding phage-encoded effector. WT phage lysates lacking effectors served as negative controls. cps major capsid protein, gp gene product, hoc highly immunogenic outer capsid protein, 168 phage K1 gene product 168, kb kilobase. Source data are provided as a Source data file.

Uropathogen Collection; Supplementary Fig. 1)[20] identified 25 different bacterial species, with *E. coli* (34%), *Enterococcus faecalis* (18%), and *Klebsiella pneumoniae* (14%) as the most prevalent uropathogens, which is consistent with previous etiological studies on UTIs[25].

Guided by these observations, we engineered HEPTs using five distinct and strictly lytic phages that target the predominant uropathogens *E. coli* (phages E2 and CM001), *E. faecalis* (phages EfS3 and EfS7), and *K. pneumoniae* (phage K1). These phages represent various phylogenetic families with distinct virion morphologies and genome sizes[20] (Fig. 1b and Supplementary Table 1). To target Gram-negative uropathogens, we selected four colicin-like bacteriocins (CLBs) as effectors, which are active against either *E. coli* (colicins E6, M, and E7; *green*) or *K. pneumoniae* (klebicin M; *purple*). CLBs are protein toxins with narrow-spectrum activity, characterized by a distinct, three-domain architecture that mediates receptor binding (central domain), Ton- or Tol-dependent translocation (N-terminal domain), and periplasmic or intracellular toxicity (C-terminal domain)[26]. The cytotoxicity of CLBs that were selected for HEPT engineering is mediated by periplasmic peptidoglycan biosynthesis inhibition (colicin M, klebicin M), intracellular 16s rRNase activity (colicin E6), or unspecific cytosolic nuclease activity (colicin E7)[26,27]. To target the Gram-positive uropathogen *E. faecalis*, we employed a phage-derived, chimeric cell wall-hydrolase (EC300; *blue*) that recognizes and degrades *E. faecalis* cell walls with high specificity[28]. EC300 was engineered by fusing an M23 endopeptidase domain from a virion-associated lysin with a cell wall binding domain of an *E. faecalis* phage endolysin[28].

All effector genes were codon-optimized to match scaffold target species specificity[29] and integrated within the phage structural gene cassette alongside a strong ribosomal binding site to guide late promoter-driven expression (see Fig. 1c–f). Overall, 14 HEPTs were constructed using five distinct phage scaffolds carrying one of five different payload genes. Engineered HEPT candidates were constructed either using CRISPR-Cas9-assisted engineering[10,20] or by rebooting synthetic genomes in suitable surrogate hosts[30]. To avoid toxicity and fitness costs during phage production, CLB-encoding HEPTs were engineered and amplified in the presence of their respective bacteriocin immunity proteins, which were constitutively expressed from an independent plasmid. The production of active effector protein upon engineered phage infection was demonstrated by spot assays using crude wildtype (WT) phage or HEPT lysates on a selection of clinical target strains (Fig. 1d–f). To ensure that phage activity does not interfere with these assays, we tested lysates from engineered phages encoding effectors with cross-genus activity. All effectors were produced and active against a broad range of urine-derived isolates of the respective target species[20] with variable levels of antibacterial activity depending on the phage scaffold, which is most likely due to differences in protein expression level and/or the lysate preparation. The activity profiles of all cross-genus HEPTs against different uropathogenic bacterial isolates can be found in Supplementary Table 2 and Supplementary Data 1. The klebicin M and EC300 encoding HEPTs targeted up to 54% and 92% of tested isolates, respectively. Among the three colicins, colicin E7 presented the broadest range of activity with HEPT EfS3::*colE7* active against 70% of the 56 urological *E. coli* isolates tested.

Polymicrobial infections are commonly observed within the urinary tract, particularly during catheter-associated UTIs[31], which may complicate therapeutic phage selection, combination, and treatment. Interestingly, analysis of the Zurich Uropathogen Collection[20] revealed 35% of UTI cases (80/227) as polymicrobial, with *E. faecalis* identified as a common co-infector associated with polymicrobial UTIs involving *E. coli* (39%) and *K. pneumoniae* (26%) (Supplementary Fig. 1c). We therefore assessed the ability of HEPTs to deliver effectors with cross-genus activity to target polymicrobial communities composed of different combinations of clinical isolates through enzymatic collateral damage (cross-targeting HEPTs, Fig. 2a). All cross-targeting HEPTs that

were tested in co-culture infection experiments are shown in Fig. 2b. Using turbidity reduction and time kill assays combined with differential plating, we demonstrate the ability of HEPTs EfS3::*colE7* and EfS7::*colE7* to infect and kill *E. faecalis* cells (P, producer) while simultaneously eradicating co-cultured *E. coli* (R, recipient) within two hours of treatment through in situ release of colicin E7 effectors (Fig. 2c, d). Similarly, we demonstrate that the klebicin M-producing HEPTs EfS3::*kvarM* and Efs7::*kvarM* can successfully control co-cultures of *E. faecalis* (P) and *K. pneumoniae* (R) (Fig. 2e, f), suggesting that CLB effectors provide a viable strategy for cross-genus targeting. Furthermore, since the protective peptidoglycan layer of Gram-positive pathogens such as *E. faecalis* is externally accessible, cell wall hydrolases are also promising enzyme antibiotics (enzybiotics[32]) for cross-genus HEPT engineering. As shown in Fig. 2g, h, we demonstrate the ability of an EC300-producing HEPT based on *E. coli* phage CM001 (CM001::*ec300*) to target *E. faecalis* in co-culture, resulting in approximately 4-log reduction in *E. faecalis* cell count after six hours of treatment for a co-culture of *E. coli* Ec20 (P) and *E. faecalis* Ef12 (R) (Fig. 2h). Nevertheless, regrowth of resistant bacteria was consistently observed over extended periods, with some rebounded to saturation in as short as 10 h, underscoring the need for additional measures to combat resistance (Fig. 2c–h).

Regardless of the importance of polymicrobial infections, most UTIs are caused by a single uropathogen, with *E. coli* and *K. pneumoniae* as predominant agents[25]. Infection of monocultures with WT phages typically leads to substantial initial host killing, as can be observed in turbidity reduction assays. However, within hours of infection, stable or transient phage resistance frequently occurs, leading to regrowth of phage resistant populations. To demonstrate this well-known limitation for phage treatment, we infected urine-derived *E. coli* (Ec20 and Ec41) or *K. pneumoniae* (Kp18, Kp28, and Kp37) isolates with WT phages E2 or K1, respectively. As expected, regrowth was observed within <18 h and a second round of infection demonstrated that these cells no longer responded to phage challenge, with both transient and stable phage resistance identified for individual clones isolated after the second round of infection (Supplementary Fig. 2).

To circumvent this limitation, HEPTs were engineered to target resistant subpopulations through phage-mediated delivery of CLBs that provide an orthogonal killing mechanism against the same target species (self-targeting HEPTs, Fig. 3a). To this end, we constructed HEPTs E2::*colE7* and K1::*kvarM* (Fig. 3b) to treat *E. coli* or *Klebsiella* monocultures and compared performance to their WT phage counterparts. Depending on the target isolate, E2 WT treatment led to rapid (*E. coli* Ec20) or delayed (*E. coli* Ec41) regrowth due to the expansion of tolerant or resistant subpopulations, respectively (Supplementary Fig. 2a, b). Strikingly, treatment with E2::*colE7* led to a sustained (18 h) reduction in optical density and dramatic reduction of *E. coli* cell counts (>6-log reduction as compared to E2 WT at 18 h post infection) (Fig. 3c). Similar results were obtained for the self-targeting HEPT K1::*kvarM*, which strongly reduced *Klebsiella* cell counts at 10 h post-infection (p.i.) and delayed the regrowth of *Klebsiella* cells when measured at 18 h p.i. (Fig. 3d).

In experimental phage therapy, polymicrobial infections would typically require simultaneous application of multiple phages targeting each infecting bacterial species (i.e., a phage cocktail)[33]. To test this approach in the context of UTI, we assessed the antimicrobial activity of wildtype and HEPT cocktails for the control of *E. coli*/*K. pneumoniae* co-cultures in synthetic urine (Fig. 3e, f). Using a combination of two self-targeting HEPTs (E2::*colE7* and K1::*kvarM*), we were able to demonstrate a strongly enhanced killing of both *E. coli* and *K. pneumoniae* in co-culture (Fig. 3e). In addition, akin to the use of dual self-targeting HEPTs, the combination of two cross-targeting HEPTs (E2::*kvarM* and K1::*colE7*) led to a comparable improvement of *E. coli* and *Klebsiella* killing in co-culture as well (Fig. 3f). Notably, in both

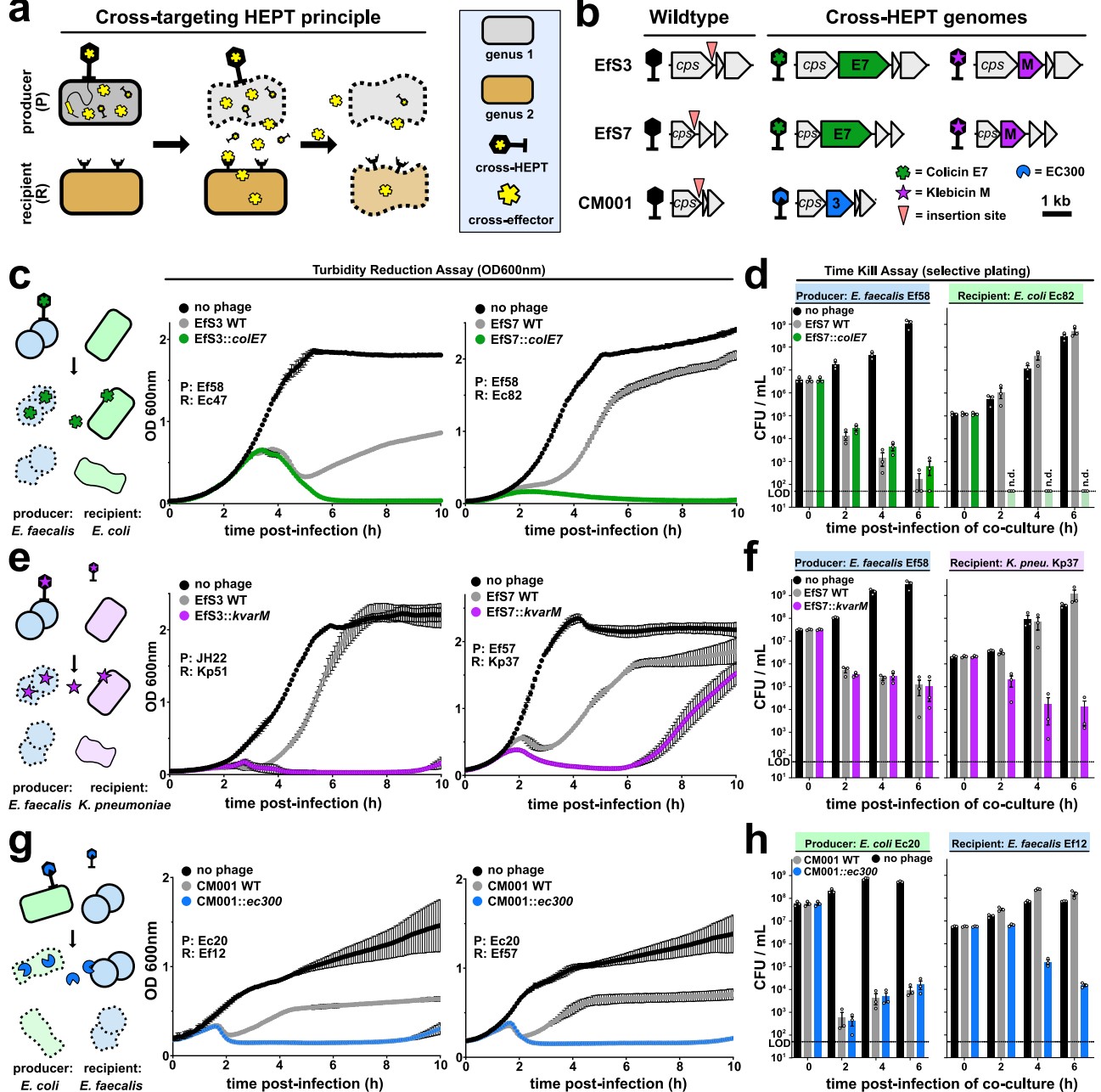

**Fig. 2 | Cross-targeting HEPTs control polymicrobial uropathogen communities. a** During cross-HEPT treatment, a phage-susceptible host produces and releases an effector with cross-genus activity that acts on bystander cells within polymicrobial communities. This approach enables polymicrobial targeting using a single phage. **b** Cross-targeting HEPTs were engineered using *Enterococcus* phages EfS3 and EfS7 and *E. coli* phage CM001 as scaffolds for effector gene insertion. **c–h** The antimicrobial effect of cross-targeting HEPTs was compared to their WT counterparts upon infecting co-cultures of a phage-susceptible host (P, producer) and an effector-susceptible target (R, recipient). Bacterial killing was quantified using 10 h turbidity reduction assays (**c**, **e**, **g**). 6 h time kill assays were combined with plating on differential and selective agar to enumerate different bacterial species (**d**, **f**, **h**). Strains used from the Zurich Uropathogen Collection: *E. faecalis* Ef12/57/58; *E. coli* Ec20/47/82; *K. pneumoniae* Kp37/51; cps, major capsid protein; LOD = limit of detection; OD$_{600nm}$ = optical density at 600 nm. Turbidity reduction data (**c**, **e**, **g**) are technical triplicates shown as mean ± SD; time kill assay data (**d**, **f**, **h**) are biological triplicates presented as mean ± SEM. Source data are provided as a Source data file.

cases, while *E. coli* Ec41 exhibited moderate sensitivity toward phage-mediated killing, *K. pneumoniae* Kp37 was highly resistant to treatment when targeted by phage alone, potentially due to its encapsulation and biofilm formation capacity (grey and green boxes). However, the emergence of bacterial resistance and consequent resurgence in cell growth was effectively mitigated by imposing secondary damage from co-released *Klebsiella*-targeting effectors (purple and red boxes). Combinations of two HEPTs outperformed cocktails containing one HEPT and one wildtype phage and the presence of any colicin E7-

producing HEPT led to complete eradication of *E. coli*. This data suggest that each bacterial species can be leveraged as an effector-producing host, even if it might not be the pathogenic agent of the polymicrobial UTI. Overall, HEPTs provide an effective, localized, and two-pronged attack to target bacteria and prevent or delay their regrowth through in situ production of self- or cross-targeting effectors.

In the future, phage-based precision antimicrobials will likely be designed and implemented as personalized treatment options.

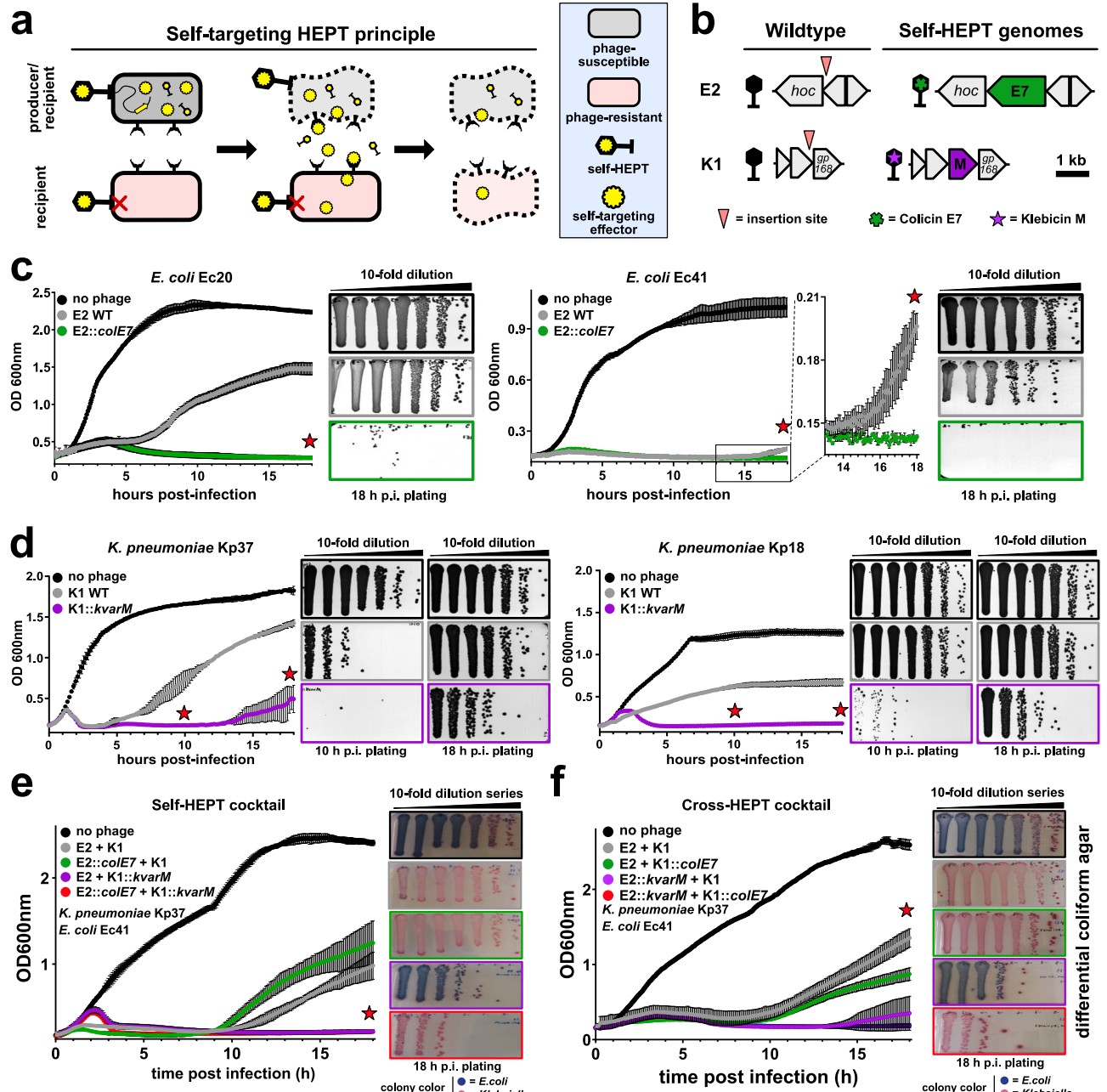

**Fig. 3 | Self-targeting HEPTs enhance killing of uropathogenic *E. coli* and *K. pneumoniae* isolates through release of colicin-like bacteriocins. a** Propagation of bacteria that survive WT phage treatment (e.g., phage-resistant subpopulations, pink) results in a failure to control bacterial growth. Self-targeting HEPTs can prevent or delay the growth of resistant subpopulations in situ by releasing complementary antimicrobial effectors, resulting in a two-pronged attack against a single bacterial target. **b** Genes encoding colicin E7 (green) or klebicin M (purple) were integrated within the structural gene cassette of phage scaffolds E2 or K1 to generate HEPTs targeting *E. coli* (E2::*colE7*) or *K. pneumoniae* (K1::*kvarM*). **c**, **d** Turbidity reduction assays combined with timepoint plating (red stars) demonstrated improved antimicrobial activity (i.e., regrowth was avoided or

delayed) for E2::*colE7* (**c**) and K1::*kvarM* (**d**) compared to WT phage treatment of uropathogenic *E. coli* and *K. pneumoniae* monocultures, respectively.
**e**, **f** Combination treatment of an *E. coli/K. pneumoniae* co-culture with a self-HEPT (**e**) or cross-HEPT (**f**) cocktail was performed. Growing cultures of *E. coli* and *K. pneumoniae* were adjusted to OD$_{600nm}$ of 0.1, mixed at a ratio of 1:1, and infected with the indicated WT phages and/or HEPTs ($5 \times 10^7$ PFU/mL). Optical density was monitored over 18 h of infection at 30 °C, followed by differential plating on chromogenic coliform agar (matching box and curve colors). hoc highly immunogenic outer capsid protein, gp gene product, OD$_{600nm}$ = optical density at 600 nm. Turbidity reduction data are technical triplicates shown as mean ± SD. Source data are provided as a Source data file.

Therefore, a rapid and reliable companion diagnostic would be helpful to guide phage selection and/or predict therapeutic success[20,34,35]. To assess the performance of self-targeting HEPTs against *E. coli* in patient urine, we combined our recently developed reporter phage-based diagnostic[20] with ex vivo urine treatment using HEPT E2::*colE7* (workflow: Fig. 4a). To this end, 39 fresh and untreated patient urine samples from the Balgrist University Hospital (Zürich, Switzerland) were

collected and immediately subjected to reporter phage-based *E. coli* identification and phage susceptibility screening using a nanoluciferase-encoding reporter phage E2 (E2::*nluc*)[20]. Reporter phage-induced bioluminescence was quantified within 5 h as an indicator of successful phage genome delivery and expression. Concurrently, the presence of *E. coli* in individual urine specimens was screened and confirmed using differential plating. Eight out of thirty-

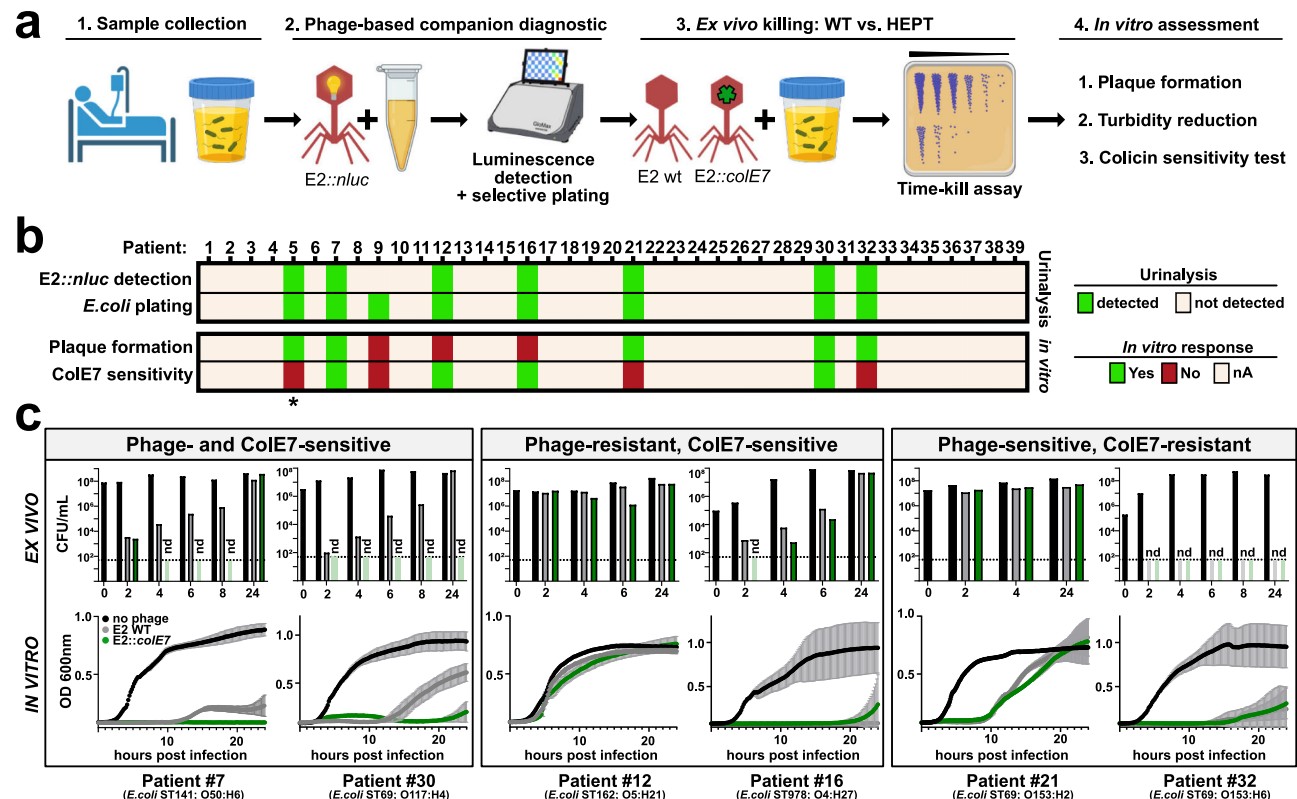

**Fig. 4 | HEPTs provide enhanced killing of colicin-sensitive *E. coli* in patient urine. a** Workflow for combining a phage-based companion diagnostic to identify potential HEPT responder patients presenting with E2-sensitive *E. coli* bacteriuria (steps 1 and 2) with subsequent ex vivo treatment (step 3) and in vitro assessment (step 4) of positive urine specimens using E2 WT or E2::*colE7*. **b** Patient urine samples (*n* = 39) were subjected to a bioluminescence-based (E2::*nluc*) reporter phage assay[20] to identify E2-sensitive *E. coli* in the urine within 4.5 h. Urine was plated on differential agar to isolate patient strains and enumerate overall bacterial load. *E. coli* isolates were further tested in vitro for E2 sensitivity (plaque formation)

and colicin E7 sensitivity (detailed results provided in Supplementary Fig. 3) and categorized: Category I = phage and colicin sensitive; Category II = phage-resistant, colicin-sensitive; Category III = phage-sensitive, colicin-resistant. **c** Time kill assays were used to assess ex vivo treatment using $10^9$ PFU/mL E2 or E2::*colE7* added to fresh urine for 24 h at 37 °C (*n* = 1). A similar HEPT treatment was performed in vitro on patient isolates grown in SHU with bacterial killing measured using turbidity reduction assays (technical triplicates shown as mean ± SD). Elements of (**a**) were created with BioRender.com. Source data are provided as a Source data file.

nine urine samples were culture-positive for *E. coli*, seven of which were detected by E2::*nluc* (Fig. 4b and Supplementary Fig. 3a, urinalysis).

To assess the advantage of heterologous effector delivery, all E2::*nluc*-responsive urine samples were subjected to HEPT (E2::*colE7*) or WT phage treatment, with *E. coli* killing quantified over 24 h using time kill assays (Fig. 4c, ex vivo). In addition, patient-derived *E. coli* strains were isolated and tested for phage and colicin E7 susceptibility, with phage susceptibility defined as the ability to form plaques (Fig. 4b and Supplementary Fig. 3b). Specimen #5 was excluded due to the presence of >$10^7$ CFU/mL *K. pneumoniae* (detected only after 16 h plating on differential agar). The remaining six patient isolates could be classified into three categories based on their susceptibility profiles: (I) phage- and colicin-sensitive, (II) phage-resistant but colicin-sensitive, and (III) phage-sensitive but colicin-resistant. Compared to E2 WT, improved *E. coli* killing by E2::*colE7* was observed for both ex vivo treatments of urine containing phage- and colicin-sensitive *E. coli* (specimens #7 and #30), which was subsequently confirmed in vitro using turbidity reduction assays performed in SHU (Fig. 4c, in vitro). For urine containing phage-resistant but colicin-sensitive *E. coli* (specimens #12 and #16), a slight enhancement of activity was observed during ex vivo HEPT treatment; however, no improvement could be validated during in vitro turbidity reduction analysis. This was attributed to incomplete infectivity of phage E2 towards specimens #12 and #16, as observed by a lack of plaque formation (Supplementary Fig. 3b). As expected, due to a lack of colicin sensitivity for specimens

#21 and #32, no difference in activity was observed between E2 WT and E2::*colE7* for urine containing only phage-sensitive *E. coli*. Overall, the ex vivo study demonstrated enhanced HEPT-mediated killing of *E. coli* in fresh patient urine, provided that the isolate is susceptible to both phage and effector, i.e., colicin E7. To further evaluate the potential benefit of E2::*colE7*, we compared it's performance versus E2 WT using an additional eleven *E. coli* isolates from the Zurich Uropathogen collection[20] and a model UTI strain (MVAST0072) with different phage and colicin susceptibility profiles (Fig. 1d and Supplementary Fig. 3d). HEPT vs. WT phage activity was assessed using turbidity reduction assays in synthetic urine, followed by endpoint plating (Supplementary Fig. 4). In accordance with the ex vivo urine treatment results presented in Fig. 4, while prominent enhancement in HEPT-mediated killing was observed with all phage- and effector-sensitive isolates, the absence of colicin E7 susceptibility rendered E2::*colE7* comparable to its wildtype counterpart. For future implementation of HEPTs and other phage-based therapeutics, careful and rapid screening of relevant susceptibility profiles, e.g., using reporter phage companion diagnostics, may become an essential component to support their therapeutic precision.

In conclusion, we present HEPTs as precision antimicrobials that combine the inherent, pathogen-specific killing activity of bacteriophages with in situ production and release of secondary antimicrobial effectors. This localized, two-pronged approach enhances the antimicrobial activity of phages, is capable of suppressing outgrowth of phage resistant subpopulations, and can be

harnessed to provide cross-genus control of bacterial pathogens using a single HEPT or HEPT cocktails. Compared to wildtype phage cocktails, recombinant effector proteins, or a combinatorial treatment involving both as separate components, HEPTs offer several advantages from developmental, therapeutic, and translational perspectives. Firstly, owing to the simplicity of our engineering pipelines and the inherent genomic plasticity of phages, a single phage scaffold can be easily engineered to carry different, and potentially multiple, effector genes, generating HEPTs capable of delivering an assortment of therapeutic effects. In addition to antimicrobial enhancement, HEPTs could also be equipped with other payload classes, e.g., immune modulators or biofilm-dispersing enzymes, to further benefit bacterial clearance and patient outcomes. Secondly, as HEPT amplification and concomitant payload release are reliant on phage infection of the target bacteria, the activity from released effectors is limited to sites of bacterial infection, which ensures for optimal temporal and spatial effector activity with minimal off-target effects. Thirdly, assisted by their genetically encoded secondary antimicrobials, HEPT(s) can achieve the desired therapeutic effects using a minimal number of phage constituents, largely reducing the effort, cost, and complexity associated with phage manufacturing, formulation, and administration. For UTI treatment, previously established intravesical instillation protocols can be adopted to deliver HEPT(s) directly to the site of infection while minimizing microbiome disturbances and the risk of systemic side effects[36]. Furthermore, the growing availability of phage scaffolds and effector libraries, coupled with advances in genome editing, functional screening, and analytical techniques, offer exciting possibilities for the establishment of a design-build-test-learn platform to rapidly customize HEPTs toward individual patient's needs. This personalized treatment approach, along with reporter phage-based companion diagnostics, holds great promise for improving antimicrobial therapy not only for UTIs but also for other infectious diseases.

## Methods

### Compliance with ethical regulations

Our research complies with all relevant ethical regulations. All patients who provided urine samples (as outlined in Fig. 4) gave a general written informed consent, in line with the local ethics committee (Kantonale Ethikkommission Zurich, Switzerland), agreeing for further use of health-related personal data and biological material for research purposes. The study was performed in accordance with the World Medical Association Declaration of Helsinki[37] and conformed with the International Conference on Harmonisation (ICH) Good Clinical Practice (GCP) Guidelines (E6) and the International Organization for Standardization (ISO, 14,155).

### Bacterial strains and culture conditions

*E. coli* BL21 (New England Biolabs), *E. coli* Ec20[20], *K. pneumoniae* KpGe[38], *E. faecalis* JH2-2, and *E. faecalis* Ef57[20] were used as phage propagation and engineering hosts. *E. coli* XL1-Blue MRF' (Stratagene) was used as a cloning host for plasmid construction. Clinical strains used in this study are listed in Supplementary Table 2 and Supplementary Data 1, which includes isolates taken from the Zurich Uropathogen Collection; a library of 663 patient isolates identified from urine specimens of patients from the Department of Neuro-Urology, Balgrist University Hospital, Zurich, Switzerland, acquired between January and December 2020 and provided after routine testing by the Institute of Medical Microbiology (IMM), University of Zurich[20]. Gram-negative bacteria were grown at 37 °C in Lysogeny Broth (LB) or Synthetic Human Urine (SHU[39]). Gram-positive bacteria were cultivated at 37 °C in BHI-fc broth (37 g/L Brain-Heart-Infusion broth from Biolife Italiana, 4 g/L glycine, 3.2 mM L-cysteine HCl, 50 mM Tris, 5 ng/mL choline chloride) under microaerophilic conditions.

### Phage propagation and purification

Phages were propagated on their respective propagation hosts using the soft-agar-overlay method as described previously[20]. To avoid toxicity, colicin-encoding HEPTs E2::*colE7*, K1::*colE7*, and K1::*colE6* were propagated in the presence of their immunity plasmids, pIm_immE7 and pIm_immE6, respectively (Supplementary Table 1 and Supplementary Data 2). In brief, after overnight incubation, phage particles were extracted using 5 ml SM buffer per plate (50 mM Tris, pH 7.4, 100 mM NaCl, 8 mM MgSO$_4$) and filter-sterilized (0.2 μm) to obtain crude phage lysates (used for effector susceptibility testing as described below). Phage lysates were further purified and concentrated by PEG precipitation (7% PEG 8000 and 1 M NaCl) followed by cesium chloride isopycnic centrifugation and finally dialyzed against a 1000-fold excess of SM buffer.

### Bacterial genome sequencing

Bacterial strains used for in vitro HEPT assessment were sequenced and typed (provided in Supplementary Data 1). Bacterial gDNA was extracted using a GenElute™ Bacterial Genomic DNA Kit (NA2120, Merck, USA) and Illumina sequenced (2 × 150 bp) by Eurofins Genomics Europe Sequencing GmbH (Constance, Germany). Contigs were de novo assembled using the CLC Genomics Workbench version 20 (QIAGEN Bioinformatics) with default settings and subsequently assessed using the multi locus sequence typing (MLST) and SeroTypeFinder services provided by the Center for Genomic Epidemiology at the Technical University of Denmark (www.genomicepidemiology.org/).

### Phage genome sequencing

*E. coli* phage CM001 was isolated from a mixture of wastewaters collected in Switzerland using *E. coli* Ec20 as a host, purified using three sequential rounds of the soft-agar-overlay method, and further propagated as described above. Genomic DNA was extracted from purified phage particles using the phenol:chloroform:isoamyl alcohol (25:24:1) extraction method. Purified DNA was Illumina sequenced (2 × 150 bp) by Eurofins Genomics Europe Sequencing GmbH (Constance, Germany). A single contig was obtained by de novo assembly using the CLC Genomics Workbench version 20 (QIAGEN Bioinformatics) with default settings. Coding DNA sequence (CDS) identification and annotation was performed using the RAST server[40], with tRNAscan-SE used to identify possible tRNA genes (none detected)[41]. Subsequent manual curation and validation was performed using related *E. coli* phage K1H (NC_027994) as a reference genome.

### Transmission electron microscopy

Phage particles were negatively stained for 20 s with 2% uranyl acetate on carbon-coated copper grids (Quantifoil) and observed at 100 kV on a Hitachi HT 7700 equipped with an AMT XR81B Peltier cooled CCD camera (8 M pixel) at the ScopeM facility, ETH Zurich.

### CRISPR-Cas9-assisted phage engineering

All HEPTs based on phages E2, K1, EfS3, and EfS7::*colM/kvarM* were constructed using the homologous recombination-based and CRISPR-Cas9-assisted engineering as previously described[20]. In short, WT phages were propagated in the presence of the respective editing template (pEdit) to enable sequence-specific transgene integration through homologous recombination. WT phages were selectively restricted using a SpyCas9-based counterselection system (pSelect) directed at the flanking homology arms within individual phage genomes. Silent mutations within the protospacer-adjacent motifs (PAMs) on the homology arms of pEdit enable CRISPR-escape and enrichment of engineered phage. When PAM mutation is impossible, multiple silent mutations were introduced within the SpyCas9-targeted seed sequence (12 nucleotides immediately upstream of the PAM) to abrogate CRISPR targeting. All pEdit and pSelect vectors used for

phage construction are compiled in Supplementary Data 2; synthetic DNA strings used for effector amplification are shown in Supplementary Data 3.

## Phage genome assembly and rebooting

EfS7::*colE7* and CM001::*ec3OO* genomes were assembled in vitro from six PCR fragments with ~40 bp overlapping ends using the Gibson isothermal method (NEBuilder HiFi DNA assembly master mix, NEB) as previously described[20]. In brief, 20 ng of PCR products per 1 kb of genomic fragment length were used for assembly. Synthetic genomes of *E. faecalis* HEPTs were rebooted through transfection into *L. monocytogenes* Rev2L L-form bacteria as previously described[30]. To reboot CM001::*ec3OO*, 3 µl of assembly mixture was electroporated into 42 µL of electrocompetent *E. coli* XL1-Blue cells at 1.8 kV, 25 µF, 200 Ω using a BTX ECM630 electroporator (BTX Molecular Delivery Systems, MA, USA). 1 mL of SOC medium was supplemented immediately after electroporation and cells were recovered at 37 °C for 4 h with shaking (180 rpm). Subsequently, 10 µl of chloroform (132950, Sigma-Aldrich) was added to assist host lysis and phage release. Following centrifugation at 12,000 x g for 1 min, dilutions of the supernatant were mixed with 200 µL of an overnight culture of *E. coli* Ec20 and 5 mL of molten LC soft agar, layered onto pre-warmed LB plates, and incubated for 16 h at 37 °C. Synthetic DNA strings used for amplification of effector genes are compiled in Supplementary Data 3. Primers and templates used for phage genome fragmentation and effector gene integration are summarized in Supplementary Table 3.

## Effector susceptibility assessment

400 µL of a log-phase culture of the target bacterial strain was mixed with 10 mL of molten LC soft agar, poured onto a square plate (12 × 12 cm) containing the appropriate growth agar, and dried for 15 min. 10 µL of each sterile-filtered crude phage lysate (-10^9–10^10 PFU/mL) was spotted on the bacterial lawn, dried, and incubated overnight. Bacterial susceptibility to effectors presented in HEPT lysate would result in the formation of growth inhibition zone.

## Turbidity reduction assays

Log-phase cultures were diluted in BHI-fc (*E. faecalis*) or SHU (*E. coli* and *K. pneumoniae*) to an $OD_{600nm}$ of 0.05 − 0.1, distributed into clear, flat-bottom 96-well plates (Bioswisstech) and infected with phages to obtain a final concentration of $5 \times 10^7$ plaque-forming units (PFU)/mL. The plates were sealed with a microplate sealing film (AxygenTM) and $OD_{600nm}$ was quantified every 5 min at 30 °C using a spectrophotometer (SPECTROstar Omega or SPECTROstar Nano, BMG Labtech). Uninfected bacterial dilutions were used as growth controls, and growth medium without bacteria was used as a background/sterility control. All cross-genus HEPT experiments used a ratio of 10:1 producer to recipient cells. Experiments were performed as technical triplicates and reported as mean ± standard deviation (SD). Data was prepared and analysed using GraphPad Prism (Version 9). When indicated, triplicate reactions were combined, serially diluted, and plated on agar plates at 10 or 18 h post-infection.

## Time kill assays (TKAs)

For cross-genus HEPT TKAs, 1 mL of co-culture was infected with $10^9$ or $5 \times 10^7$ PFU/mL of HEPTs derived from EfS7 and CM001 scaffolds, respectively. The ratio of producer cell to recipient cell was always 10:1 with starting concentrations (CFU/mL) provided in Fig. 2. Differential and selective plating was used to enumerate the different bacterial species. *E. coli* and *K. pneumoniae* were counted on chromogenic coliform agar (CCA, Biolife italiana) and *E. faecalis* was counted on KFS agar (Biolife italiana) after incubation at 37 or 42 °C for 16 or 48 h, respectively. Experiments were performed as biological triplicates and reported as mean ± SEM. Data was prepared and analysed using GraphPad Prism (Version 9).

## Repeated phage exposure and resistance development

*E. coli* strains Ec20 and Ec41 and *K. pneumoniae* strains Kp18, Kp28, and Kp37 were diluted to an $OD_{600nm}$ of 0.1 and infected with $10^8$ PFU/mL of phages E2 or K1 (round I) in SHU. After 18 h of infection, phage-exposed bacterial cultures were combined ($n = 3$), diluted to an $OD_{600nm}$ of 0.1 in fresh SHU, and spiked with the WT phage ($10^8$ PFU/mL) or fresh media again for another 18 h (round II). Individual surviving clones were isolated after round II (36 h total experimental time) by re-streaking on LB agar and tested in vitro for phage susceptibility using spot-on-the-lawn assays.

## Reporter-phage-based urinalysis and ex vivo activity assessment

Reporter phage urinalysis was performed as described previously[20]. In brief, 1 mL of patient urine sample was directly mixed with 4 mL of LB. Samples were enriched for 1 h at 37 °C with shaking (180 rpm). 50 µL of reporter phage was added to individual 450 µL aliquots of enriched urine ($10^6$ PFU/mL final concentration) and incubated at 37 °C with shaking. LB media spiked with reporter phages alone served as background controls. Bioluminescence measurements were taken at 3 h post infection. Based on the manufacturer's instructions, a buffer-reconstituted NLuc substrate (Nano-Glo Luciferase Assay System, Promega) was mixed at a 1:1 ratio with a sample of the infection mixture (40 µL total) in Nunc™ F96 Micro-Well™ plates (Thermo Fisher). Bioluminescence was quantified 5 min after substrate addition using a GloMax® Navigator Luminometer (Promega) with 5 s integration and 2 s delay. Relative light units (RLUs) were background corrected by division of the RLU from phage-only controls. Samples producing $>10^3$ RLU fold change (FC) were considered positive. To confirm reporter phage results and to isolate strains, patient urine was plated on differential agar (UriSelect4, BioRad). *E. coli* isolates were tested for phage-sensitivity by determining the efficiency of plating using the soft-agar-overlay method. For ex vivo TKA experiments, 250 µL of patient urine was infected with 250 µL of a $10^9$ PFU/mL stock of E2 or E2::*colE7* in PBS solution, incubated for 24 hours at 37 °C, and plated on LB at the indicated time points. T = 0 was plated prior to phage addition as an input control. To replicate TKA conditions, in vitro turbidity reduction assays were performed by adding 100 µl of a $10^9$ PFU/mL stock solution of E2 or E2::*colE7* to 100 µl of bacterial culture in SHU at a lower starting $OD_{600nm}$ of 0.005 to reproduce bacterial loads in patient urine.

## Statistics and reproducibility

No statistical method was used to predetermine sample size. A single transmission electron microscopy image of each phage (Fig. 1b) was selected from multiple micrographs taken from a single phage-coated grid as is common practice. All TKAs are biological triplicates excluding the ex vivo urine treatment. The field evaluation, including bioluminescence-based detection and ex vivo TKA, was performed once due to time and material limitations of working with multiple fresh urine samples. Phage susceptibility test (efficiency of plating; Supplementary Fig. 3b) was performed in biological triplicates with isolated strains. All TRA (and corresponding endpoint CFU plating) and spot lysis data were reproduced at least two times, with one set of data shown.

## Reporting summary

Further information on research design is available in the Nature Portfolio Reporting Summary linked to this article.

## Data availability

The phage CM001 genome is available from the GenBank database (UTI-CM001, OM810255) alongside previously sequenced genomes of phage E2 (OL870316), K1 (OL870318), EfS3 (OL870611), and EfS7 (OL870612). Original data used to analyze UTI incidences

(Supplementary Fig. 1) were sourced from Meile et al.[20]. Source data are provided with this paper.

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

## Acknowledgements

We would like to thank all members of the CAUTIphage consortium including Jochen Klumpp, Hendrik Koliwer-Brandl, Jonas Marschall, Shawna McCallin, Vera Neumeier, and Reinhard Zbinden for their

scientific input and assistance with organizing clinical samples. We thank the clinical care team at the Department of Neuro-Urology, Balgrist University Hospital (Zurich, Switzerland) for urine specimen collection. We also thank Karin Moelling for scientific discussions, Alexander Harms for providing *E. coli* UTI89 and CFT073, and Leo Meile for providing *E. faecalis* JH2-2. Finally, we acknowledge the ScopeM facility at ETH Zurich for assistance with transmission electron microscopy of phage particles. J.D., L.L., T.M.K., M.J.L., and M.D. were supported by a Sinergia grant (CRSII5_189957) from the Swiss National Science Foundation (SNSF). S.M. and S.K. were supported by an Ambizione grant (PZOOP3_174108) from the SNSF.

## Author contributions

Conceptualization, J.D., S.M., M.J.L., S.K., M.D.; methodology, J.D., S.M., S.K., M.D.; project administration, S.K., M.D.; supervision, S.K., M.D.; investigation, J.D., S.M., J.B., T.J., P.P., L.H., C.I.M., S.K., M.D.; data curation, J.D., S.M., S.K., M.D.; visualization, S.K.; writing—original draft, S.K., M.D.; writing—review and editing, J.D., S.M., L.L., T.M.K., M.J.L., S.K., M.D.; funding acquisition, L.L., T.M.K., M.J.L., S.K., M.D.; resources, T.M.K., M.J.L., S.K.

## Competing interests

M.D. and S.K. are also employees of Micreos GmbH and M.J.L. is a scientific advisor to Micreos GmbH. The other authors declare no competing interests.
