## [Peer review file · Nature Communications]

REVIEWER COMMENTS

Reviewer #1 (Remarks to the Author):

In this manuscript the authors insert genes with antimicrobial potential (colicins and lysin) into different phages genomes in order to enhance their antibacterial activity. Using this strategy, the authors are able to simultaneously target two different bacterial populations that the native phage can't do. Although the study is interesting and experimentally well designed, the concept is not really new. There are already a number of studies focused on improving phages' activity through the addition of antimicrobial payloads (e.g. enzymes) or to target different bacterial populations (some refs: Lu et al, 2007; Pei et al, 2014; Ando et al, 2015; Yosef et al. 2017).

General concerns related with the manuscript:

- 5 lytic phages were used and 4 colicins and 1 lysin. However, it is not clear the selection of effector proteins to be cloned into phages. While in both *E. faecalis* phages the same 3 colicins were cloned, in *E. coli* phages different effector proteins were inserted: in phage E2 were inserted 3 effector proteins while in phage CM001, only the lysin was inserted (Fig. 1b) - Why? Also, why the E6 was only cloned into K1 and not the others?
- Why the selection of Efs3:colE7 and Efs7:colE7 for the infection experiments (Fig 2 c) and not the others? Was this based on the best range of activity?
- How do you explain the differences in *E. coli* activity between the 2 Enterococcus phages carrying the same bacteriocin (Table S2)
- Fig. 2: I would like to see the turbidity reduction assay for 24h and also the CFU numbers at this time point
- Some problems with references 19 and 33
- Not clear the experimental design of the resistance experiment (Line 354). How many rounds with phage were performed?
- In fig. 4c the correlation between ex vivo and in vitro is not clear. For instance, the grey line in patient #7 at 24h seems to be approximately 0.3 and the number of CFUs close to 10^8 , the same happens with the green line in patient #16. However, in patient #32 both green and grey lines are around the same OD as the previous ones mentioned but no CFUs were detected?

Reviewer #2 (Remarks to the Author):

The study “Enhancing bacteriophage therapeutics through in situ production and release of heterologous antimicrobial effectors” describe the design and testing of synthetically modified phages called heterologous effector phage therapeutics (HEPTs) to eradicate UTI pathogens, especially in polymicrobial infections. HEPTs were prepared by sophisticated protocols to augment lytic phage genomes with colicin-like bacteriocins and cell wall hydrolases.

It was shown that HEPTs suppress resistance and improve uropathogen killing by dual phage- and effector-mediated targeting. Moreover, the authors designed the E2::nluc reporter phage for companion diagnostics to identify potential HEPT responder patients.

There are some major comments to this study:

1. The authors proposed heterologous effector phage therapeutics (HEPTs) as a model system of precision antimicrobials against UTI pathogens, but there was no discussion provided about the possible delivery of HEPTs to reach targeted bacteria. If oral administration, how do the authors envision the influence of HEPTs on gut flora?
2. What is the bacterial receptor recognized by phages selected for HEPT design? Is this receptor a conservative structure present in the majority of strains?
3. What were the host range and EOP of WT phages compared to their self-targeting HEPTs tested on available UTI strain collection? It would be useful to predict the HEPTs propagation efficiency on commonly isolated UTI strains compared to WT phages. Moreover, it would show the applicability of the E2::nluc reporter phage in general.
4. Bacterial 10-fold dilution and phage dilution scales should be reversed on the charts, from the smallest to the largest (left to right).
5. Figure 2c-f, please explain why different producers were used for Efs3 and Efs7 phages in each experiment.
6. Please provide in Figure 2 the turbidity reduction and killing assay results for 18h experiments.
7. Figure 3e-f. Please explain no differences in the colony reduction for E2+K1 versus E2::colE7+K1 and E2+ K1::colE7. In the growth kinetics, the self-HEPT cocktail was even less effective than WT phage cocktails.
8. Please discuss why the HEPTs are superior to WT phage cocktails or WT phage combinations with bacteriocins and cell wall hydrolases as recombinant proteins.
9. Please discuss the advantages and limitations of recombinant bacteriocins or cell wall hydrolases application versus self-targeting HEPTs.

Reviewer #3 (Remarks to the Author):

This manuscript presents an approach for engineering phages to be more effective for therapeutic applications. The authors insert genes encoding various bactericidal toxins into the genomes of phages. These “heterologous effector phage therapeutics” (HEPTs) are shown to be more effective at eradicating targeted bacterial strains associated with urinary tract infections. Importantly, they show that the HEPTs more effectively eliminate a strain compared to the wild-type phage by greatly reducing the frequency of resistant strains arising. They also show that HEPTs can replicate in one species and produce toxins that will kill completely unrelated species. This could be a useful property for infections caused by multiple bacterial species. The authors show that phages expressing luciferase can be used to rapidly identify strains that are sensitive to a given phage and that these strains can be subsequently eradicated using a HEPT. Some of these experiments are performed using fresh untreated urine, demonstrating the potential for practical clinical use of these engineered phages in treating urinary tract infections.

Overall, this is a very interesting paper that warrants publication in Nature Communications. I have some minor suggestions to improve the clarity of the paper as listed below.

Line 117- The authors should add more detail about how the phage engineering was done. Although they have published these some approaches previously, a few diagrams in the supplementary figures to show exactly what was done here would be useful. This would include both the phages engineered with CRISPR and those that were synthesized. For the synthesis, I would like to see how many fragments were assembled in the Gibson reaction and the identity of these fragments.

Line 120- The authors should make it clear that the immunity protein was expressed from a plasmid the strain used to produce the phages. I was confused about this at first as I thought the immunity protein was on the phage, which would not be useful for further use.

In Figures 1d,e, and f, it is difficult to figure out what is being shown. The authors should explain the assay in the figure legend. If this is phage being spotted on a bacterial lawn, what are the white spots? Is this a negative picture? What is the concentration of phage spotted? I am a phage biologist and I found these figures difficult.

Table S2: What is the concentration (pfu/ml) of the phage lysate used for the experiment

Fig. S2: What “stable” and “transient” mean? How were these properties determined?

Enhancing bacteriophage therapeutics through *in situ* production and release of heterologous antimicrobial effectors

Reviewer response

Comments from Reviewer #1:

In this manuscript the authors insert genes with antimicrobial potential (colicins and lysin) into different phages genomes in order to enhance their antibacterial activity. Using this strategy, the authors are able to simultaneously target two different bacterial populations that the native phage can't do. Although the study is interesting and experimentally well designed, the concept is not really new. There are already a number of studies focused on improving phages' activity through the addition of antimicrobial payloads (e.g. enzymes) or to target different bacterial populations (some refs: Lu et al, 2007; Pei et al, 2014; Ando et al, 2015; Yosef et al. 2017).

General concerns related with the manuscript:

1. - 5 lytic phages were used and 4 colicins and 1 lysin. However, it is not clear the selection of effector proteins to be cloned into phages. While in both *E. faecalis* phages the same 3 colicins were cloned, in *E. coli* phages different effector proteins were inserted: in phage E2 were inserted 3 effector proteins while in phage CM001, only the lysin was inserted (Fig. 1b) - Why? Also, why the E6 was only cloned into K1 and not the others?

The overall aim of our manuscript was to introduce and promote the concept of heterologous effector phage therapeutics (HEPTs), where distinct phage scaffolds can be equipped with diverse effectors for cross-genus targeting or enhanced host killing. Overall, we have developed fourteen HEPTs using five scaffolds carrying one of five candidate effector genes and have demonstrated their functionality both as monophage or polyphage cocktails *in vitro* and *ex vivo*. While we acknowledge that the construction of additional HEPTs (e.g., CM001::*colE6*) could potentially improve the manuscript, we believe that our results already provided a strong and comprehensive proof of concept for this technology.

2.- Why the selection of EfS3:*colE7* and EfS7:*colE7* for the infection experiments (Fig 2 c) and not the others? Was this based on the best range of activity?

As mentioned on line 137 and presented in Fig. S1, *E. faecalis* was identified as the most common co-infecter associated with polymicrobial UTIs involving *E. coli* and *K. pneumoniae*. We therefore used infection curves from cross-targeting *Enterococcus* phages (EfS3/EfS7::*colE7* and Efs3/Efs7::*kvarM*) as representations to demonstrate the cross-genus killing concept of HEPTs. Further data about cross-HEPTs can also be found in Fig. 3f, where we used K1::*colE7* (*Klebsiella* phage expressing colicin) and E2::*kvarM* (coliphage expressing klebicin) as cocktail to control *E. coli* and *K. pneumoniae* co-cultures. Besides, since *colE7* was identified as the most promising colicin with the broadest range of activity (line 132, Table S2), *colE7*-based HEPTs were selected for further *in vitro* and *ex vivo* experiments.

3.- How do you explain the differences in *E. coli* activity between the 2 *Enterococcus* phages carrying the same bacteriocin (Table S2)

Since the heterologous bacteriocin genes were placed under the control of phage native promoters, their expression levels can vary depending on individual promoter strength. Furthermore, differences in phage lysate preparation may lead to differences in payload yield between phages. Accordingly, we have modified our text (line 128) to explain the variable levels of antibacterial activity observed between different phage scaffolds: "All effectors were produced and active against a broad range of urine-derived isolates of the respective target species (19) with variable levels of antibacterial activity depending on the phage scaffold, which is most likely due to differences in protein expression level and/or the lysate preparation"

4.- Fig. 2: I would like to see the turbidity reduction assay for 24h and also the CFU numbers at this time point.

The goal of cross-HEPT is to tackle polymicrobial communities within a short timeframe instead of addressing the issue of phage resistance as we later performed using the self-targeting HEPTs where a longer duration of activity, i.e., 18 hours (shown in Fig. 3) and 24 hours (shown in Figs. 4 & S4), is desired and was subsequently assessed.

Nevertheless, we appreciate the reviewer's suggestion, and therefore to avoid misunderstanding of the cross-targeting HEPT data, we have added the following statement on Line 156: "*Nevertheless, regrowth of resistant bacteria was consistently observed over extended periods, with some rebounded to saturation in as short as 10 h, underscoring the need for additional measures to combat resistance (Fig. 2c-h).*"

5.- Some problems with references 19 and 33

Thank you for bringing this to our attention. Both references have been corrected.

6.- Not clear the experimental design of the resistance experiment (Line 354). How many rounds with phage were performed?

In total two consecutive rounds of phage infection were performed to demonstrate bacterial resistance development. Text has been improved in the method section (line 386) to clarify the experimental design: "*E. coli strains Ec20 and Ec41 and K. pneumoniae strains Kp18, Kp28, and Kp37 were diluted to an OD_{600nm} of 0.1 and infected with 10⁸ PFU/mL of phages E2 or K1 (round I) in SHU. After 18 h of infection, phage-exposed bacterial cultures were combined (n=3), diluted to an OD_{600nm} of 0.1 in fresh SHU, and spiked with the WT phage (10⁸ PFU/mL) or fresh media again for another 18 h (round II). Individual surviving clones were isolated after round II (36 h total experimental time) by re-streaking on LB agar and tested in vitro for phage susceptibility using spot-on-the-lawn assays.*"

7.- In fig. 4c the correlation between ex vivo and in vitro is not clear. For instance, the grey line in patient #7 at 24h seems to be approximately 0.3 and the number of CFUs close to 10⁸, the same happens with the green line in patient #16. However, in patient #32 both green and grey lines are around the same OD as the previous ones mentioned but no CFUs were detected?"

All *ex vivo* treatments were performed by direct phage addition into patient urine, whereas all follow-up *in vitro* experiments were performed with pure cultures in synthetic human urine. Although similar trends were observed under both conditions (i.e., HEPTs provide enhanced killing of colicin-sensitive *E. coli*), it is not unexpected that the complex urine matrices have additional influences on phage activity, potentially due to variations in bacterial amount and viability, pH, ionic strength, and the presence of immune factors or antibiotics when compared to a synthetic environment. Therefore, one can expect the results from these two separate experiments to be correlated but not always the same (as pointed out with the patient #32 data).

Nevertheless, we agree this could be better explained and as such we have modified the text (line 227) to clarify the differences in experimental setup between the *ex vivo* and *in vitro* experiments: "*Compared to E2 WT, improved E. coli killing by E2::colE7 was observed for both ex vivo treatments of urine containing phage- and colicin-sensitive E. coli (specimens #7 and #30), which was subsequently confirmed in vitro using turbidity reduction assays performed in SHU (Fig. 4c, in vitro).*"

Comments from Reviewer #2:

The study “Enhancing bacteriophage therapeutics through in situ production and release of heterologous antimicrobial effectors” describe the design and testing of synthetically modified phages called heterologous effector phage therapeutics (HEPTs) to eradicate UTI pathogens, especially in polymicrobial infections. HEPTs were prepared by sophisticated protocols to augment lytic phage genomes with colicin-like bacteriocins and cell wall hydrolases.

It was shown that HEPTs suppress resistance and improve uropathogen killing by dual phage- and effector-mediated targeting. Moreover, the authors designed the E2::nluc reporter phage for companion diagnostics to identify potential HEPT responder patients.

There are some major comments to this study:

1. The authors proposed heterologous effector phage therapeutics (HEPTs) as a model system of precision antimicrobials against UTI pathogens, but there was no discussion provided about the possible delivery of HEPTs to reach targeted bacteria. If oral administration, how do the authors envision the influence of HEPTs on gut flora?

Thank you for the suggestion. We have added the following to the concluding section regarding a potential administration route for UTI treatment using HEPTs (line 266): “*For UTI treatment, previously established intravesical instillation protocols can be adopted to deliver HEPT(s) directly to the site of infection while minimizing microbiome disturbances and the risk of systemic side effects (Leitner et al., 2021).*”

Reference: Leitner, L. *et al.* Intravesical bacteriophages for treating urinary tract infections in patients undergoing transurethral resection of the prostate: a randomised, placebo-controlled, double-blind clinical trial. *Lancet Infect. Dis.* 21, 427–436 (2021).

2. What is the bacterial receptor recognized by phages selected for HEPT design? Is this receptor a conservative structure present in the majority of strains?

Thank you for the question. While we are certain that our selected HEPT scaffolds can infect broad ranges of uropathogens (based on previous host range analysis [Meile et al., 2022]), their bacterial receptors are still currently being investigated.

Reference: Meile, S., Du, J., Staubli, S., Grossmann, S., Koliwer-Brandl, H., Piffaretti, P., Leitner, L., Matter, C.I., Baggenstos, J., Hunold, L. et al. (2022) Engineered reporter phages for rapid detection of *Escherichia coli*, *Klebsiella* spp., and *Enterococcus* spp. in urine. *bioRxiv*, 2022.2011.2023.517494.

3. What were the host range and EOP of WT phages compared to their self-targeting HEPTs tested on available UTI strain collection? It would be useful to predict the HEPTs propagation efficiency on commonly isolated UTI strains compared to WT phages. Moreover, it would show the applicability of the E2::nluc reporter phage in general.

Since phage host ranges are primarily determined by their receptor binding proteins, which remained untouched during HEPT construction, we expect no tropism difference between self-targeting HEPTs and their parental WT phages. Besides, we believe propagation efficiency is not an essential parameter to evaluate the performance of HEPTs, because good killer phages for antimicrobial application should enable rapid and sustained clearance of their bacterial hosts, which in turn might restrict their own propagation. On the other hand, mediocre killer phages that propagate efficiently and equilibrate with their bacterial hosts are not suitable for biocontrol. In other words, high propagation efficiency does not necessarily mean good treatment outcome. A better parameter to evaluate HEPTs performance would be their killing efficacy. As demonstrated both *ex vivo* and *in vitro* in Fig. 4, Fig. S3, and Fig S4, the killing efficacy of HEPTs on uropathogens can be reliably predicted based upon phage- and effector-

susceptibility test, where enhanced killing with HEPTs is expected on all strains showing high susceptibility to both. Moreover, we believe that phage propagation efficiency is irrelevant to the applicability of E2::*nluc* reporter phages, because luminescence generation from E2::*nluc* relies solely on phage genome injection and luciferase expression. Results concerning the transduction range and infection kinetics of E2::*nluc* can be found in our separate and cited manuscript (Meile et al., 2022, under review).

Reference:

Meile, S., Du, J., Staubli, S., Grossmann, S., Koliwer-Brandl, H., Piffaretti, P., Leitner, L., Matter, C.I., Baggenstos, J., Hunold, L. et al. (2022) Engineered reporter phages for rapid detection of *Escherichia coli*, *Klebsiella* spp., and *Enterococcus* spp. in urine. *bioRxiv*, 2022.2011.2023.517494.

4. Bacterial 10-fold dilution and phage dilution scales should be reversed on the charts, from the smallest to the largest (left to right).

Thank you for spotting this. We have corrected the dilution scales in Fig 3, Fig S2, and Fig S4.

5. Figure 2c-f, please explain why different producers were used for Efs3 and Efs7 phages in each experiment.

As mentioned in the main text (line 139): “*We therefore assessed the ability of HEPTs to deliver effectors with cross-genus activity to target polymicrobial communities composed of different combinations of clinical isolates through enzymatic collateral damage*”. We intentionally selected different producer-recipient combinations to demonstrate the wide applicability of HEPTs in clinical settings, where a diverse array of strains is encountered every day.

6. Please provide in Figure 2 the turbidity reduction and killing assay results for 18h experiments.

The goal of cross-HEPT is to tackle polymicrobial communities within a short timeframe instead of addressing the issue of phage resistance development as we later performed using the self-targeting HEPTs where a longer duration of activity, i.e., 18 hours (Fig. 3) and 24 hours (Figs. 4 & S3), is desired and was subsequently assessed.

Nevertheless, we appreciate the reviewer’s suggestion and as such to avoid misinterpretation of the cross-targeting HEPT data we have added the following statement on Line 156, “*Nevertheless, regrowth of resistant bacteria was consistently observed over extended periods, with some rebounded to saturation in as short as 10 h, underscoring the need for additional measures to combat resistance (Fig. 2c-h).*”

7. Figure 3e-f. Please explain no differences in the colony reduction for E2+K1 versus E2::*colE7*+K1 and E2+ K1::*colE7*. In the growth kinetics, the self-HEPT cocktail was even less effective than WT phage cocktails.

Thank you for the suggestion. We have included additional text on line 190 to explain no differences in the colony reduction for E2+K1 versus E2::*colE7*+K1 and E2+K1::*colE7*: “*Notably, in both cases, while E. coli Ec41 exhibited moderate sensitivity toward phage-mediated killing, K. pneumoniae Kp37 was highly resistant to treatment when targeted by phage alone, potentially due to its encapsulation and biofilm formation capacity (grey and green boxes). However, the emergence of bacterial resistance and consequent resurgence in cell growth was effectively mitigated by imposing secondary damage from co-released Klebsiella-targeting effectors (purple and red boxes).*”

8. Please discuss why the HEPTs are superior to WT phage cocktails or WT phage combinations with bacteriocins and cell wall hydrolases as recombinant proteins.

Answered below.

9. Please discuss the advantages and limitations of recombinant bacteriocins or cell wall hydrolases application versus self-targeting HEPTs.

Thank you for the above two suggestions, to outline the advantages of HEPT(s), we have added the following text to the conclusion section (line 252): “Compared to wildtype phage cocktails, recombinant effector proteins, or a combinatorial treatment involving both as separate components, HEPTs offer several advantages from developmental, therapeutic, and translational perspectives. Firstly, owing to the simplicity of our engineering pipelines and the inherent genomic plasticity of phages, a single phage scaffold can be easily engineered to carry different, and potentially multiple, effector genes, generating HEPTs capable of delivering an assortment of therapeutic effects. In addition to antimicrobial enhancement, HEPTs could also be equipped with other payload classes, e.g., immune modulators or biofilm-dispersing enzymes, to further benefit bacterial clearance and patient outcomes. Secondly, as HEPT amplification and concomitant payload release are reliant on phage infection of the target bacteria, the activity from released effectors is limited to sites of bacterial infection, which ensures for optimal temporal and spatial effector activity with minimal off-target effects. Thirdly, assisted by their genetically encoded secondary antimicrobials, HEPT(s) can achieve the desired therapeutic effects using a minimal number of phage constituents, largely reducing the effort, cost, and complexity associated with phage manufacturing, formulation, and administration. For UTI treatment, previously established intravesical instillation protocols can be adopted to deliver HEPT(s) directly to the site of infection while minimizing microbiome disturbances and the risk of systemic side effects (Leitner et al., 2021). Furthermore, the growing availability of phage scaffolds and effector libraries, coupled with advances in genome editing, functional screening, and analytical techniques, offer exciting possibilities for the establishment of a design-build-test-learn platform to rapidly customize HEPTs toward individual patient’s needs. This personalized treatment approach, along with reporter phage-based companion diagnostics, holds great promise for improving antimicrobial therapy not only for UTIs but also for other infectious diseases.”

Reference: Leitner, L. *et al.* Intravesical bacteriophages for treating urinary tract infections in patients undergoing transurethral resection of the prostate: a randomised, placebo-controlled, double-blind clinical trial. *Lancet Infect. Dis.* 21, 427–436 (2021).

Comments from Reviewer #3:

This manuscript presents an approach for engineering phages to be more effective for therapeutic applications. The authors insert genes encoding various bactericidal toxins into the genomes of phages. These “heterologous effector phage therapeutics” (HEPTs) are shown to be more effective at eradicating targeted bacterial strains associated with urinary tract infections. Importantly, they show that the HEPTs more effectively eliminate a strain compared to the wild-type phage by greatly reducing the frequency of resistant strains arising. They also show that HEPTs can replicate in one species and produce toxins that will kill completely unrelated species. This could be a useful property for infections caused by multiple bacterial species. The authors show that phages expressing luciferase can be used to rapidly identify strains that are sensitive to a given phage and that these strains can be subsequently eradicated using a HEPT. Some of these experiments are performed using fresh untreated urine, demonstrating the potential for practical clinical use of these engineered phages in treating urinary tract infections.

Overall, this is a very interesting paper that warrants publication in Nature Communications. I have some minor suggestions to improve the clarity of the paper as listed below.

1. Line 117- The authors should add more detail about how the phage engineering was done. Although they have published these some approaches previously, a few diagrams in the supplementary figures to show exactly what was done here would be useful. This would include both the phages engineered with CRISPR and those that were synthesized. For the synthesis, I would like to see how many fragments were assembled in the Gibson reaction and the identity of these fragments.

Thank you for the suggestion. The CRISPR engineering pipeline used here has been described before, and furthermore, is detailed for the specific phages used here in our associated manuscript (Meile et al., 2022; currently under review and being prepared for back-to-back publication), so we believe it would be redundant to include another diagram here. Regarding the synthetic engineering pipeline, methods have been extensively described in our previous publications (Kilcher et al, 2018). Nevertheless, to clarify the experimental design, we have modified our text in the method section (line 343): “*Efs7::colE7 and CM001::ec300 genomes were assembled in vitro from six PCR fragments with ~40 bp overlapping ends using the Gibson isothermal method (NEBuilder HiFi DNA assembly master mix, NEB) as previously described.*” Additionally, lengths of individual fragments used for synthetic phage construction have been added to the supplementary table (Table S6).

References:

Meile, S., Du, J., Staubli, S., Grossmann, S., Koliwer-Brandl, H., Piffaretti, P., Leitner, L., Matter, C.I., Baggenstos, J., Hunold, L. et al. (2022) Engineered reporter phages for rapid detection of *Escherichia coli*, *Klebsiella* spp., and *Enterococcus* spp. in urine. bioRxiv, 2022.2011.2023.517494.

Kilcher, S. et al. (2018) Cross-genus rebooting of custom-made, synthetic bacteriophage genomes in L-form bacteria. Proc. Natl. Acad. Sci. U. S. A. 115, 567–572.

2. Line 120- The authors should make it clear that the immunity protein was expressed from a plasmid the strain used to produce the phages. I was confused about this at first as I thought the immunity protein was on the phage, which would not be useful for further use.

Thank you for the suggestion. On line 120, we have modified the text to: “*To avoid toxicity and fitness costs during phage production, CLB-encoding HEPTs were engineered and amplified*

in the presence of their respective bacteriocin immunity proteins, which were constitutively expressed from an independent plasmid.”

3. In Figures 1d,e, and f, it is difficult to figure out what is being shown. The authors should explain the assay in the figure legend. If this is phage being spotted on a bacterial lawn, what are the white spots? Is this a negative picture? What is the concentration of phage spotted? I am a phage biologist and I found these figures difficult.

Thank you for the suggestion. We have modified our text in the legend of Fig. 1 to include phage concentration and additional explanation on the clear zones (white spots): *“(Fig. 1d-f) Cross-genus antimicrobial activity of crude WT phage or HEPT lysates ($\sim 10^9$ - 10^{10} PFU/mL) was tested against clinical uropathogen isolates using the spot-on-the-lawn method (full lists provided in Tables S2 and S3). Clear zone formation at the site of HEPT lysate spot indicates bacterial susceptibility to the corresponding phage-encoded effector. WT phage lysates lacking effectors served as negative controls.”*

4. Table S2: What is the concentration (pfu/ml) of the phage lysate used for the experiment

Thank you for the suggestion. We have added the concentration of phage lysates in the heading of Table S2 and S3: *“10 μ L of phage lysate (10^9 - 10^{10} PFU/mL) was spotted on bacterial lawns of the following isolates with activity assessed visually after 16 h incubation at 37°C.”*, and in the method section (line 360): *“10 μ L of each sterile-filtered crude phage lysate ($\sim 10^9$ - 10^{10} PFU/mL) was spotted on the bacterial lawn, dried, and incubated overnight.”*

5. Fig. S2: What “stable” and “transient” mean? How were these properties determined?

Thank you for the question. We have added the following text in the Fig. S2 legend to explain the difference between “stable” and “transient” resistance: *“After the second round of infection, individual clonal survivors were isolated, purified via three consecutive rounds of streaking, and assessed for phage susceptibility using spot-on-the-lawn assays. Progenies of bacterial survivors that retained resistance to phage plaquing were considered as stably resistant, whereas those that regained permissiveness to phage plaquing were rendered transiently resistant. Non-infected clones served as positive controls.”*

REVIEWERS' COMMENTS

Reviewer #1 (Remarks to the Author):

Although in some cases the authors were not very clear and objective in their answers (comments 1 and 7), the manuscript significantly improved and all the comments/concerns raised were addressed.

There is still one important question that should be answered before publication: are the phages carrying the effector genes stable after several rounds of propagation? It would be important to perform an experiment to address this

Enhancing bacteriophage therapeutics through *in situ* production and release of heterologous antimicrobial effectors

Reviewer response

Reviewer #1 (Remarks to the Author):

Although in some cases the authors were not very clear and objective in their answers (comments 1 and 7), the manuscript significantly improved and all the comments/concerns raised were addressed. There is still one important question that should be answered before publication: are the phages carrying the effector genes stable after several rounds of propagation? It would be important to perform an experiment to address this.

We sincerely appreciate the reviewer's valuable time spent on reviewing our manuscript again, and for their suggestion to further investigate our heterologous effector phage therapeutic (HEPT) phages through a genetic stability assessment. We acknowledge the importance of assessing genetic stability for any clinical utility of our HEPT phages. However, we respectfully disagree that this type of data is essential to our current study or to demonstrate proof-of-concept for the HEPT principle.

Regarding our original responses to comments 1 and 7, we would like to emphasize that the primary goal of this study was to present (i) the engineering principle for producing HEPTs, (ii) their activity as self-targeting and cross-genus targeting HEPT, and (iii) their improved antibacterial activity compared to wildtype phages for potential treatment of bacterial UTIs. Our aim was to demonstrate that distinct and diverse phage scaffolds can be equipped with effectors (bacteriocins and endolysins) for cross-genus targeting and enhanced host killing. It is important to clarify that these HEPTs were not all constructed simultaneously by our team but were developed using various phage scaffolds and different payloads in an iterative manner, allowing us to continuously adapt our choice of scaffolds and explore new ideas. Ultimately, we present data for fourteen individual HEPTs and further investigate the utility of E2::*colE7* *ex vivo* as it exhibited the most effective and broad-range self-targeting HEPT characteristics.

Regarding our response to comment 7, we want to emphasize that the *ex vivo* and *in vitro* data presented in Figure 4 represent separate experiments. The *ex vivo* data was generated directly using urine samples, while the *in vitro* data was obtained by culturing pure cultures of the *E. coli* strain isolated from the patient's urine in synthetic human urine. Although similar trends were observed in both conditions, it is reasonable to expect differences due to the complex nature of urine matrices, including variations in bacterial amount and viability, pH, ionic strength, and the presence of immune factors or antibiotics, which may influence phage activity. Therefore, while the results from these separate experiments may correlate, they may not always be identical (as detailed in the main text for the patient #32 data). We believe we have adequately addressed this issue in our previous revised submission of the manuscript.

Once again, we sincerely thank the reviewer for their valuable feedback, and we hope that our responses adequately address their concerns.